# Development and Validation of the Safety Behavior Assessment Form-PTSD Scale

**DOI:** 10.3390/bs15091248

**Published:** 2025-09-12

**Authors:** Jason T. Goodson, Madison E. Fraizer, Gerald J. Haeffel, Jacek Brewczynski, Lucas Baker, Caleb Woolston, Anu Asnaani, Erika M. Roberge

**Affiliations:** 1VA Salt Lake City Healthcare Systems, PTSD Clinical Team, Building 16, 500 Foothill Boulevard, Salt Lake City, UT 84148, USA; 2Department of Psychology, College of Arts and Letters, University of Notre Dame, Notre Dame, IN 46556, USA; mfraize3@nd.edu (M.E.F.);; 3Department of Psychology, College of Social and Behavioral Science, University of Utah, Salt Lake City, UT 84112, USA

**Keywords:** safety behaviors, PTSD, anxiety, measurement, treatment monitoring, reliability, validity

## Abstract

Safety behaviors are mental processes and behaviors associated with the onset, maintenance, and treatment of anxiety-related disorders. But these behaviors are understudied in the context of posttraumatic stress disorder (PTSD). One reason is the lack of psychometrically valid instruments to assess safety behaviors specific to the diagnosis of PTSD. To address this gap in the literature, we adapted a well-validated general measure of safety behaviors to create a brief 10-item questionnaire for assessing PTSD-specific safety behaviors—the Safety Behavior Assessment Form-PTSD scale (SBAF-PTSD scale). The results of four studies, using both clinical and non-clinical populations, supported the reliability and validity of the SBAF-PTSD scale; the measure demonstrated strong internal consistency, test–retest reliability, inter-item correlations, and convergent and divergent validity across all four studies. It also demonstrated clinical utility as it predicted treatment outcomes for American military veterans diagnosed with PTSD. Results provide initial support for this measure as a tool that can be used in both research and in clinical practice (e.g., treatment monitoring).

## 1. Introduction

Safety behaviors are actions and mental processes that people engage in to avoid feared outcomes ([29]). They are expressed in a variety of forms ranging from reassurance seeking to carrying medications, to social evasion and escape. Although many safety behaviors are idiosyncratic, different disorders tend to have prototypical safety behaviors. For example, safety behaviors in social anxiety tend to include self and social vigilance related to fears of negative evaluation ([27]) whereas in generalized anxiety, they often take the form of repetitive information seeking and strategies related to minimizing uncertainty ([1]; [6]). In panic disorder, safety behaviors tend to center around bodily vigilance and arousal reduction strategies ([30]) and in trauma and PTSD, safety behaviors tend to focus on situational vigilance, suppression of trauma memories, and avoidance related to internal and external threats (e.g., scoping places before entering, sitting with one’s back to a wall, checking locks on doors and windows, walking slowly to allow someone walking behind to pass; [10]; [11]).

There is strong evidence that safety behavior use is associated with negative mental health outcomes. Individuals who engage in greater levels of safety behaviors are more likely to develop anxiety ([12]; [31]) and have worse treatment outcomes ([15]; [3]) than those who do not engage in these behaviors. Most of the research on safety behaviors has focused on those related to social anxiety disorder ([13]). Far fewer studies have investigated safety behaviors associated with PTSD specifically (see [10]; [12] for exceptions). This is a problem because safety behaviors have strong theoretical and empirical links to PTSD. Research indicates that safety behavior use precedes and predicts increases in PTSD symptoms ([13]). This makes sense because PTSD tends to be characterized by an ongoing sense of threat, and safety behavior use can reinforce beliefs about threat because they facilitate avoidance, which prevents the individual from learning the probability of the feared outcome is low (i.e., prevents extinction). In addition, there is a need for a PTSD-specific measure of safety behaviors to use for measurement-based care. A brief measure is needed for treatment monitoring that can track progress on the reduction in PTSD-specific safety behaviors as these are often resistant to change in evidence-based treatments for PTSD ([8]).

One reason for the lack of research on PTSD-specific safety behaviors, despite their importance, is the absence of measurement options ([14]). Most measures of safety behaviors are limited in scope, focusing mainly on social anxiety safety behaviors ([13]). This led [13] ([13]) to create the Safety Behavior Assessment Form (SBAF), which is a broad measure of safety behaviors relevant to generalized anxiety disorder, panic disorder, social anxiety, health anxiety, and PTSD. The measure has been used in over 20 studies using multiple independent samples differing on clinical severity of symptoms (clinically significant psychopathology, moderate levels, and “healthy controls”), age (undergraduates and middle-aged adults from the community and VA hospitals), geographic regions (North America, Western and Eastern Europe), and races and ethnicities. It also has been tested in a variety of designs including cross-sectional, longitudinal, and prevention and treatment interventions. In these studies, the SBAF has demonstrated strong reliability metrics (both internal consistency and test–retest reliability), convergent and divergent validity (e.g., it differentiates clinical and non-clinical participants; see [13]), and predictive validity in longitudinal designs (e.g., predicts PTSD treatment outcomes and increases in anxiety in non-clinical samples).

However, despite demonstrating strong psychometric properties in prior work, the SBAF does not fill the gap in the literature for a measure of PTSD-specific safety behaviors. That is because most of the items on the SBAF are not relevant to PTSD ([4]). In addition, the SBAF is 41 items, making it too long to use in real-world clinical settings, especially if it were to be used for weekly treatment monitoring.

The purpose of this research was to revise the SBAF to create a new stand-alone measure of PTSD-specific safety behaviors that is reliable, valid, and brief. To this end, we conducted four studies. The purpose of Study 1 was to guide the final item selection for the SBAF-PTSD scale by testing its factor structure and psychometric properties. The purpose of Studies 2 and 3 were to replicate the results found in Study 1 in two independent samples and provide further validation (e.g., by evaluating reliability and convergent and discriminant validity) of the measure. And the purpose of Study 4 was to test the clinical utility of the SBAF-PTSD in a PTSD treatment effectiveness study.

## 2. Scale Development

### 2.1. Overview

We followed generally accepted guidelines for scale construction and validation ([7]). First, we identified a need for a measure (in this case, there is not a short, validated measure of PTSD-specific safety behaviors that could be used in research or clinical settings). Next, there was item selection. In this case, we adapted a previously published measure of safety behaviors to create the SBAF-PTSD scale (see below). Then, following an iterative design process, we evaluated the factor structure of the new scale and item fit (Study 1). This led us to remove two items from the scale. Then, we tested the structure again (Study 2) and reliability and validity. Then using two additional studies (Studies 3 and 4) we further tested reliability (internal consistency), convergent and discriminant validity, incremental validity, and predictive validity. We also provided a much-needed test of potential clinical usefulness.

### 2.2. Preliminary Scale Development-SBAF-PTSD Scale (12 Item Version)

The initial SBAF-PTSD scale was created by selecting items from the SBAF (full form). We chose to adapt an existing measure rather than start from scratch for several reasons. First, the safety behavior items used on the SBAF have already been tested in prior research; the items on the scale have already demonstrated strong psychometric properties using a variety of designs and samples as conducted by multiple research teams. Creating an entirely new item pool would likely be redundant and have less empirical support. Second, the SBAF contains a previously validated subscale labeled “vigilance” (in addition to two other subscales—health and social). Prior work (e.g., [13]) shows the vigilance subscale is internally consistent (Cronbach’s α > 0.7) and has strong test–retest reliability (>0.8). Further, the vigilance subscale has items that are theoretically relevant to PTSD, shown to be correlated with PTSD symptoms, and predictive of anxiety treatment outcomes ([13]).

Thus, to create the SBAF-PTSD scale, we first selected the seven items from the vigilance subscale of the SBAF. As already noted, these items are conceptually relevant to PTSD and have been shown to be correlated with PTSD symptoms in prior research. However, to ensure that we were not missing other kinds of safety behaviors associated with PTSD, we also considered the other items on the SBAF full scale. Specifically, we identified items that correlated with symptoms of PTSD as measured with the PCL-5. To this end, we examined correlations between the 41 items on the SBAF and PCL-5 total in two unpublished clinic data samples (gathered as part of standard routine care). The first clinic sample was 115 treatment-seeking veterans at the SLC VA PTSD Clinical Team, and the second sample was 63 veterans who completed Prolonged Exposure Therapy (PE) at the Philadelphia VA Medical Center. The analyses found that in addition to the seven items on the vigilance scale, five other items significantly and reliably correlated with the PCL-5 in both clinical samples (ranging from 0.21 to 0.48). Thus, 12 items were initially selected to comprise the SBAF-PTSD scale. It is this version of the scale that was evaluated in Study 1.

## 3. Study 1

### 3.1. Study 1—Method

#### 3.1.1. Overview

The purpose of Study 1 was to develop and validate a measure of safety behaviors specific to PTSD based on a sample of 12 items adapted from the SBAF-41 (see section titled “Preliminary Scale Development” above for how initial items were selected). Based on prior theory regarding various classifications of the safety behaviors (see Table 1), we speculated that either a one- or two-dimensional structure underlies the PTSD-specific safety behaviors. We expected that the scores on the measure would be associated with scores on a measure of PTSD symptoms to a greater extent than to a measure of depression, which would provide evidence for convergent and discriminant validity. Finally, we expected the scores on this measure to be largely independent of age, sex/gender, and race/ethnicity.

#### 3.1.2. Participants

Participants were 173 treatment-seeking veterans who presented for an assessment at a mid-sized, western United States VA PTSD clinic. Of the 173 participants, 4 did not complete all the items on the SBAF-PTSD scale and were excluded from analyses. Thus, the final sample consisted of 169 veterans. The mean age of the participants was 42.81 (*SD* = 12.47), and the majority were male (*n* = 145, 85.8%). Seventy-four percent (*n* = 125) self-identified their race/ethnicity as White, 6% (*n* = 10) as Asian, 5.3% (*n* = 9) as Black, 5.3% (*n* = 9) as Hispanic, 1% (*n* = 3) as Pacific Islander, and 8% (*n* = 13) as “other” (e.g., mixed race/ethnicity).

Sixty percent of participants (*n* = 101) were diagnosed with DSM-5 PTSD, 20% (*n* = 33) were diagnosed as “other trauma/stressor disorders”, and 18% (*n* = 30) met criteria for a non-PTSD DSM-5 diagnosis (e.g., major depressive disorder, generalized anxiety disorder). Five participants (2.0%) were not assigned a DSM-5 diagnosis. The diagnosis of other trauma/stressor disorder was given when a participant did not meet full diagnostic criteria for DSM-5 PTSD but still reported symptoms of a clinically significant level that caused distress/life interference. Diagnoses were made using the CAPS-5, which is a 30-item structured diagnostic interview. The CAPS-5 is a validated interview for making current and lifetime diagnosis of PTSD. It was administered by licensed doctoral-level clinical psychologists (and their supervisees) in the Department of Veterans affairs (National Center for PTSD).

The final sample size of 169 is acceptable for CFA considering guidelines suggesting the need for at least 10 participants per scale variable (10 participants × 12 items = 120 participants; [21]; [26]; [32]).

#### 3.1.3. Measures

*Patient Health Questionnaire-9* (*PHQ-9*; [23]). The PHQ-9 is a 9-item self-report measure designed to assess symptoms of depression. Each item on this measure corresponds to a DSM-IV diagnostic criterion for a major depressive episode. Respondents rated the frequency with which they experience each of the symptoms of depression on a 4-point scale (0 = *not at all*; 3 = *nearly every day*). The PHQ-9 has well-established psychometric properties with diagnostic validity and high levels of sensitivity and specificity for major depression ([23]). Individual items were summed to provide a total score, with higher scores indicative of greater depression severity (10–14 = *moderate*; 15–19 = *moderately severe*; 20–27 = *severe*). Cronbach’s alpha is typically greater than 0.85 in treatment-seeking veteran samples (e.g., [16]).

*PTSD Checklist—5th Edition* (*PCL-5 Past Month*; [33]). The PCL-5 is a 20-item self-report of PTSD symptoms. Respondents rate how much they were bothered by each of the 20 diagnostic symptoms of PTSD on a 5-point scale (0 = *not at all*; 4 = *extremely*) over the past month. Individual items are summed to provide a total score ranging from 0 to 80. Total scores of 33 and above represent a positive screening for PTSD ([5]). The PCL-5 has demonstrated good psychometric properties, including strong internal consistency (α = 0.91–0.95) in treatment-seeking military service members ([35]).

*SBAF-PTSD Scale (12-items).* See description under heading “preliminary scale development.”

#### 3.1.4. Data Analytic Procedures

Demographic characteristics, correlations, and ANOVAs were calculated using SPSS version 27.00 (SPSS, Chicago, IL, USA). Correlations and ANOVAs were used to compare relationships and mean differences between measures to assess convergent and divergent validity of the SBAF-PTSD scale (description of scale development above) with other constructs. Before conducting statistical analyses, data quality was confirmed, and there was no evidence that data were not-missing at random (NMAR); consequently, variable means were imputed for NMAR values. Additionally, assumptions of each statistical test to be run were checked. Critical outliers were not observed, and statistical assumptions were confirmed. However, there was evidence of non-normal multivariate distribution (e.g., significant Mardia coefficient of 24.65 for the SBAF-PTSD).

The factor structure of the SBAF-PTSD scale was examined using confirmatory factor analysis (CFA) via structural equation modeling with Analysis of Moment Structures (AMOS, version 29) software. Maximum likelihood estimation was used to produce all parameter estimates (e.g., factor loadings, correlations, errors). Bootstrap analysis was also used to assess the stability/robustness of model parameters, particularly because the item data were found to be non-normal. In all cases, 1000 bootstrap samples were used. Modification indices (MIs) were examined to evaluate the potential improvement in fit for the proposed models. Model fit for the CFAs was initially evaluated by examining the significance of all estimated parameters (regression weights, scale, error variances, and scale correlations). The overall goodness-of-fit was then evaluated with multiple fit indexes, and 90% confidence interval (CI) was used to evaluate the fit. A well-fitting model was defined as minimum non-significant chi-square (*χ*^2^), normed chi-square (*χ*^2^*/df*) < 3.00, RMSEA < 0.08, GFI, AGFI, TLI, and CFI > 0.90 (for an adequate fit), and non-significant chi-square (*χ*^2^), normed chi-square (*χ*^2^*/df*) < 2.00, RMSEA < 0.06, and various indices > 0.95 (for a good fit; e.g., [17]; [18]; [20]).

### 3.2. Study 1—Results

#### 3.2.1. Scale Composition of the SBAF-PTSD

Several confirmatory maximum likelihood analyses of the SBAF-PTSD items were completed first to evaluate the latent structure of the SBAF-PTSD based on prior theoretical considerations. As shown in Table 1, there were several item designations for the SBAF-41 (original survey) based on tests of its factor structure (see [13]; [12]); they included items categorized as being related to vigilance and social situations. And there were safety behaviors (items) categorized as being preventative (to prevent or avoid the occurrence of anxiety) or restorative (to reduce anxiety after onset). Considering this prior work, six rationally derived models were specified and evaluated for goodness of fit: (1) a one-factor model with all 12 items serving as observed reflective indicators of a unitary latent SBAF-PTSD construct; (2) a correlated two-factor model wherein all items are assigned to factors based upon prior factor assignment of the original SBAF as either “Preventative” or “Restorative”; (3) a hierarchical model wherein all items were assigned to the two first order factors of “Preventative” and “Restorative” as in previous model and then these factors were specified as reflective latent indicators of a unitary higher order SBAF-PTSD construct; (4) a correlated two-factor model wherein all items except item 3 were assigned to factors based upon prior factor assignment as either “Vigilance” or “Social”; (5) a hierarchical model wherein all items save item 3 were assigned to the two first order factors of “Vigilance” and “Social” as in previous model and then these factors were specified as reflective latent indicators of a unitary higher order SBAF-PTSD construct; and (6) a one-factor model with item 3 removed, given no prior assignment of this item on the “Vigilance” vs. “Social” dimension.

For the above models, all model parameters (i.e., factor loadings and errors) and all the factor correlations were statistically significant; however, model fit indices were poor for all tested models except for Model 4, for which there was mixed/marginal support. That said, examination of the modification indices (MIs) indicated that the models could be respecified to improve fit. Based on this, several additional CFAs of the SBAF-PTSD items were completed to evaluate the latent structure of the instrument based on rational and empirical considerations. Specifically, three respecified models were evaluated for goodness of fit: (1) a one-factor model with all items and correlated errors for items 9, 10, and 11; (2) a one-factor model with correlated errors for items 9, 10, and 11 and item 3 excluded; and (3) a one-factor model with correlated errors as described above except for items 3 and 12. This one-factor model with 10 items (removing 3 and 12) composed the final SBAF-PTSD scale (coefficient alpha = 0.84; see Appendix A for final scale with instructions).

#### 3.2.2. Examination of Model Fit Indices Indicated Best Fit for Model 3

Chi-square statistic was not significant (*χ*^2^*_(32)_* = 43.06, *p* = 0.09), normed chi-square was below 2.0 (*χ*^2^*/df* = 1.35) and GFI (0.95), TLI (0.97), and CFI (0.979) were above 0.95, indicating excellent fit. RMSEA was below 0.08, including its confidence interval (0.045; 90% CI: 0.00–0.078). While a one-factor model with correlated errors based on 10 items fits the data very well, the inclusion of the correlated errors does not help guide how to best score and interpret the test. As well, examination of the error variances suggests presence of a secondary construct underlying the SBAF-PTSD. Consequently, two bi-factor models were evaluated in the final step: (1) a bi-variate factor model with two item groupings (items 1 through 7 and items 9 through 11), and (2) a unitary factor with all 10 items assigned to a primary factor and items 9, 10, and 11 assigned to the secondary latent construct (see Figure 1).

The bi-variate factor model did not fit the data at all. The unitary factor with a secondary latent factor produced excellent fit to the data. All model parameters are statistically significant (*p* < 0.001) and all model fit indices reflect excellent model fit. In fact, the model fit indices are the same as they were for the one-factor model with correlated errors excluding items 3 and 12 (see above). In essence, the bi-factor model is equivalent to the model with correlated errors. One benefit of this model over the model with correlated errors, however, is that it provides for a justification for including items 9, 10, and 11 (items from the Social Subscale of the original SBAF) in the computation of a total SBAF-PTSD score. These three social items hang together theoretically as they measure safety behaviors used to escape and evade social situations (see Table 1). Thus, we labeled this three-item factor the SBAF-PTSD Social Escape Index (coefficient alpha = 0.80).

#### 3.2.3. Correlations with Demographic Variables

Product-moment correlation coefficients and the *χ*^2^ statistic between the 10-item SBAF-PTSD scale (as determined by CFA) and age, gender (coded as males vs. females), and ethnicity/race (coded as Caucasians vs. others) were calculated to assess the influence of the demographic variables on the SBAF-PTSD scale scores. No association was found between the SBAF-PTSD scale and age (*r* = −0.03, *p* = 0.661), gender (*χ*^2^*_(28_*_)_ = 34.23, *p* = 0.193), or race/ethnicity (*χ*^2^*_(26_*_)_ = 35.10, *p* = 0.11). Additionally, one-way ANOVAs with planned contrasts were computed to compare average SBAF-PTSD scale total scores between diagnostic categories (i.e., PTSD, other trauma/stressor-related disorder, non-trauma/stressor-related disorder). Due to violation of the homoscedasticity assumption, coefficients without equal variances assumed are presented. Consistent with expectations, participants with PTSD and other trauma/stressor-related disorders reported similar SBAF-PTSD scale scores (*M =* 26.49, *SD* = 5.80 and *M =* 24.47, *SD =* 7.65, respectively; *t =* 2.02, *p =* 0.17) that were significantly higher than those with a non-trauma/stressor-related disorder diagnosis (*M* = 22.65, *SD =* 8.10; *t =* 3.85, *p =* 0.018).

#### 3.2.4. Validity

Product-moment correlation coefficients between the SBAF-PTSD scale items and the PHQ-9 and the PCL-5 can be found in Table 2. For scale-level analyses, examination of zero-order correlations indicates that the SBAF-PTSD scale demonstrates strong relationships with the PCL Total scale and Intrusions and Arousal scales, and moderate associations with the PHQ and Avoidance and Cognitions and Mood scales. Conversely, the SBAF-PTSD Social Escape Index shows strong relationships with both the PHQ and the PCL Total scale and the opposite pattern of relationships with the remaining PCL scales (i.e., strong in magnitude relationships with Cognition and Mood and Arousal scales and moderate associations with Intrusions and Avoidance scales). Closer inspection of zero-order correlations between the SBAF-PTSD scale items and scales reveals that all items produce moderate associations with at least two of the scales and show statistically significant correlations with the remaining scales with two exceptions. Items 2 and 8 produced non-significant correlations with PHQ and Cognitions and Mood scale. Conversely, items 9, 10, and 11, which comprise the SBAF-PTSD Social Escape Index, produced moderate to strong relationships with all the scales. This suggests that items 9, 10, and 11 each are a strong indicator of safety behaviors in general and should be retained as part of the revised SBAF-PTSD scale.

### 3.3. Study 1—Discussion

The purpose of Study 1 was to test a 12-item SBAF-PTSD scale. The 12 items were adapted from an established and well-validated measure of safety behaviors (see “preliminary scale development” above). As part of a reiterative development process, we conducted several CFAs to evaluate latent structure of this 12-item SBAF-PTSD scale using an independent sample (i.e., different than the one used to create the scale). Results showed that two items could be removed, which led to a final version of the SBAF-PTSD scale with 10 items, reflecting a unitary safety behavior trait and a secondary latent construct labeled Social Escape Index; it is this 10-item SBAF-PTSD scale that was tested in Studies 2, 3, and 4 below (see Appendix A for final scale). As expected, scores on the scale were found to be unrelated to various demographic variables, including age, gender/sex, and ethnicity/race. Finally, we found that the SBAF-PTSD scale was more strongly related to PTSD than to depressive symptoms.

The SBAF-PTSD scale also demonstrated an expected pattern of relationships with depressive- and PTSD-like symptomology in correlational and regression analyses producing strong relationships with trauma-related intrusions, avoidance, and arousal. Moreover, higher scores on SBAF-PTSD reliably differentiated between those with diagnosed PTSD and/or trauma-related disorder from those without the diagnoses. Together, these results provide preliminary evidence for convergent and discriminant validity of the scale.

An interesting finding from Study 1 was the identification of the small secondary factor, made up of three items that assess safety behaviors used to escape and evade anxiety in social situations. We labeled this 3-item group the SBAF-PTSD Social Escape Index. In contrast to the SBAF-PTSD scale, the scores on the Social Escape Index were most associated with various indices of mood and cognition and behavioral avoidance. Thus, the Social Escape Index may provide clinically useful information about the presence of co-occurring depressive symptoms.

## 4. Study 2

### 4.1. Study 2—Method

#### 4.1.1. Overview

The purpose of Study 2 was to further test the structure and psychometric properties of the SBAF-PTSD scale in a second independent, non-clinical sample. First, we predicted that CFA would replicate the one-dimensional factor structure found in Study 1, and that it would demonstrate strong internal consistency and inter-item correlations. Second, we predicted that the scale would demonstrate convergent and divergent validity. Specifically, it would be more strongly correlated with measures related to cognitive aspects of anxiety (intrusive thoughts and repetitive thinking) than it would with a general measure of well-being.

#### 4.1.2. Participants

Participants were 142 unselected undergraduates (48% female) from a private midsized university in the Midwestern United States. Three participants did not complete all the study measures and were removed. Thus, the final sample consisted of 139 participants. Specific data regarding ethnicity were not collected due to an error in the questionnaire software asking for demographic information; the sample likely reflects the diversity of the university undergraduates more generally: 60% White, 17% Latino or Hispanic, 12% Asian, 10% African American, and 1% Native American.

The sample size of 142 is acceptable for CFA considering guidelines suggesting at least 10 participants per scale variable (10 participants × 10 items = 100 participants; [21]; [26]; [32]).

#### 4.1.3. Measures

*SBAF-PTSD Scale*. The SBAF-PTSD scale is a 10-item self-report questionnaire assessing PTSD-specific safety behaviors (see description of scale development above). Participants rate the frequency with which they engage in each safety behavior on a 4-point scale (0 = *never*; 3 = *always*), with scores ranging from 0 to 30 (higher scores indicating more PTSD safety behavior usage). The SBAF-PTSD scale usually takes less than a minute to complete. Items include safety behaviors related to situational vigilance (e.g., “watch others for signs of danger”), checking (e.g., “check locks on doors or windows), threat-prevention strategies (e.g., “sit with back to wall”), and interpersonal escape/evasion strategies (e.g., “leave events early”, “cut conversations short”).

*Behavioral Health Questionnaire–20* (*BHQ-20*; [22]). The BHQ-20 is a measure of global mental health, with subscales specifically evaluating well-being and life functioning. The well-being subscale consists of three items that assess distress, life satisfaction, and energy/motivation (Cronbach’s alpha 0.71; [22]). The life functioning subscale consists of four items assessing work/school, intimate relationships, social relationships, and overall life enjoyment (alpha 0.80). Each item on these two measures is rated on a 5-point scale (0 = *terrible*; 4 = *very well*), with higher scores indicative of higher levels of quality of life. Well-being scores range from 0 to 12, while life functioning scales range from 0 to 16.

*Intrusive Memory Questionnaire-5* (*IMQ-5*; [2]). The IMQ-5 is a 5-item self-report questionnaire that assesses intrusive memories. One of the symptoms of DSM-5 PTSD is “recurrent, involuntary, and intrusive distressing memories of the traumatic event(s).” This measure assesses the severity and impact of these kinds of memory problems. To this end, participants rate the frequency and severity in which they have experienced intrusive memories in the past week on a 4-point scale (0 = none or not at all; 3 = daily or severely). Total scores range from 0 to 15, with scores over 5 indicating clinically significant mental intrusions.

*Repetitive Thinking Questionnaire-10* (*RTQ-10*; [24]). The RTQ-10 is a self-report measure of perseverative thinking (i.e., rumination). Rumination is considered a transdiagnostic risk factor that is associated with a variety of mental health problems, including PTSD. Repetitive thinking/rumination is associated with PTSD (i.e., part of the nomological net of constructs that could be tested for construct validity), and thus, hypothesized to also be associated with PTSD-specific safety behaviors. Items are rated on a 5-point scale (1 = not at all true; 5 = very true). The RTQ-10 has demonstrated high internal reliability (Cronbach’s alpha ≥ 0.89), as well as convergent validity with related emotions such as anxiety, depression, and general distress.

#### 4.1.4. Procedure

Participants provided informed consent and then completed the SBAF-PTSD scale, Behavioral Health Questionnaire-20, the Intrusive Measures Questionnaire-5, and the Repetitive Thinking Questionnaire-10. This research was approved by the University of Notre Dame Human Subjects Review Board, and all guidelines and procedures were followed throughout the research process. Participants were compensated with extra credit for completing the study.

#### 4.1.5. Data Analytic Procedures

Data was collected via an online survey administered on Qualtrics. Confirmatory factor analysis, reliability analysis, and correlations were analyzed on JASP Version 0.19.3 (JASP, Amsterdam, NE). CFA was conducted to assess the scale composition of the SBAF-PTSD. Correlations were calculated to compare the SBAF-PTSD to measures of related constructs.

### 4.2. Study 2—Results

#### 4.2.1. Scale Composition

CFA of the full scale showed good fit—CFI (0.934) and TLI (0.912) are above 0.90 and RMSEA is below 0.8, including its confidence interval (0.073; 90% CI: 0.040–0.104). The mean inter-item correlation was 0.33, suggesting that items are correlated but not repetitive.

#### 4.2.2. Reliability

Reliability analysis showed a Cronbach’s alpha of 0.83, which indicates good internal consistency.

#### 4.2.3. Validity

Correlations between the SBAF-PTSD and IMQ-5, RTQ-10, and the BHQ-20 can be found in Figure 2. As predicted, the SBAF-PTSD scale was moderately positively correlated with the measures related to cognitive aspects of anxiety—IMQ-5 and RTQ-10, respectively. Conversely, the SBAF-PTSD was weakly correlated with a general measure of well-being (BHQ-20).

### 4.3. Study 2—Discussion

Study 2 provided further evidence for the reliability of the SBAF-PTSD scale, as well as its convergent and divergent validity. Correlational analysis revealed a moderate positive correlation between the SBAF-PTSD scale and the IMQ-5 and RTQ-10. These findings align with the prediction that scores on the SBAF-PTSD scale would be related to common cognitive symptoms of PTSD, such as intrusive thoughts and repetitive thinking. The moderate correlation demonstrates that the SBAF-PTSD scale is related to these measures but not redundant.

Additionally, the SBAF-PTSD scale demonstrated negative correlations with quality-of-life measures, such that higher scores on the SBAF-PTSD scale were associated with lower levels of quality of life. This finding replicates other findings showing that higher safety behavior use is associated with lower quality of life ([19]).

## 5. Study 3

### 5.1. Study 3—Method

#### 5.1.1. Overview

The purpose of Study 3 was to further test the psychometric properties of the SBAF-PTSD scale in a second independent, non-clinical sample. We predicted that the scale would demonstrate strong internal consistency and acceptable inter-item correlations. Further, we hypothesized that the measure could differentiate participants with and without trauma exposure.

#### 5.1.2. Participants

Participants (*N* = 150) for this experiment were a convenience sample as they were recruited as part of a larger study conducted by a research laboratory at a public university in the Mountain West region of the United States. They ranged in age from 18 to 41 years (M = 20.5 years, SD = 7.2), with 74.5% identifying as female and 25.5% identifying as male. The sample included participants identifying ethnically as 62.0% White, 19.7% Hispanic or Latino, 8.3% Asian/Pacific Islander, 1% Native American, 2.2% Black, and 7.3% other races. Demographics additionally included 24.2% first-generation college students and 34.6% upper income-class participants. Recruitment for this study was done via SONA, a centralized online recruitment program for recruitment of undergraduate students, for which participation in this study was awarded course credit.

#### 5.1.3. Measures

***SBAF-PTSD Scale.*** See description from Study 2.

***The Primary Care PTSD Screen For DSM-5* (*PC-PTSD-5*; [28]).** The PC-PTSD-5 is a five-item screening measure designed for use in primary care settings. The first question assesses trauma exposure (yes or no), with 5 follow-up questions assessing PTSD symptoms (i.e., nightmares, intrusions, arousal, numbing/detachment, guilt/blame). All item responses are dichotomous (yes or no) with an estimated clinical cutoff of 3 (“yes” responses on the 5 symptom questions). The PC-PTSD-5 has shown good psychometric properties, including discriminant validity (differentiating PTSD from non-PTSD respondents), test–retest reliability of 0.83, and internal reliability of 0.83 ([28]; [34]).

#### 5.1.4. Procedure

The study was completed in a single session. After informed consent was received, participants filled out the SBAF-PTSD and PC-PTSD-5 measures. Institutional Review Board approval was given for all research and procedures performed in this study. Signed informed consent was received from all participants and all data were deidentified.

#### 5.1.5. Data Analytic Procedures

One-way Analysis of Variance (ANOVA) was used to test if the SBAF-PTSD scale could significantly predict PTSD group status (as determined by the PC-PTSD-5).

### 5.2. Study 3—Results

Participants were grouped into trauma exposure groups based on their answers on the PC-PTSD-5. On this measure, the first question asks about trauma exposure. Participants who answered “no” to this question were assigned to the no trauma exposure group (*n* = 77). Participants who responded “yes” to the first question were assigned to one of two groups: Group 2 (trauma exposure with minimal PTSD symptoms) comprised individuals who endorsed *two or less* PTSD symptoms (*n* = 45); and Group 3 (trauma exposed with likely PTSD) comprised individuals who endorsed *3 or more* PTSD symptoms (*n* = 24). The cutoff of 3 or more symptoms is consistent with the recommendations of [28] ([28]), which showed that the optimal sensitivity score for the PC-PTSD scale was 3 or above.

#### 5.2.1. Reliability

Analysis of internal consistency showed a Cronbach’s alpha of 0.73, which indicates adequate internal consistency. The average inter-item correlation for the SBAF-PTSD was 0.24, which is also acceptable ([7]).

#### 5.2.2. Validity

ANOVAs were conducted to test if the PTSD-SBAF scale could differentiate participants based on their PTSD group status (see Table 3 for means). Results showed that the probable PTSD group had significantly higher SBAF-PTSD scores than no trauma (*F*_(1)_ = 5.3, *p* < 0.02), and trauma-minimal symptoms (*F*_(1)_ = 5.1, *p* < 0.02) groups. There were no significant differences between the no-trauma and trauma-minimal symptoms groups. The same pattern of results occurred when SBAF-PTSD Social Escape Index was used instead of the full scale.

### 5.3. Study 3—Discussion

The results of Study 3 provided further support for the reliability and the validity of the SBAF-PTSD scale. The SBAF-PTSD scale showed good internal consistency as well as acceptable inter-item correlations. Further, we found that the SBAF-PTSD accurately differentiated individuals with probable PTSD from those with no trauma exposure and trauma exposure and minimal symptoms. Taken together, findings from these samples support the usefulness of the SBAF-PTSD scale in non-clinical samples, thus suggesting it can be used effectively in research settings.

## 6. Study 4

### 6.1. Study 4—Method

#### 6.1.1. Overview

The purpose of Study 4 was to investigate the clinical utility of the SBAF-PTSD scale using an effectiveness study design. The study was carried out on a PTSD Clinical Team with a sample of 66 treatment completers. All treatment was delivered according to standard care on a specialized PCT Team. This study had two primary aims. First, to investigate the clinical utility of the SBAF-PTSD in a PTSD treatment study. Second, to assess changes in hypervigilance across treatment by regularly administering the SBAF-PTSD. Recent studies have found arousal in general, and hypervigilance specifically, to respond less to traditional EBPs for PTSD ([8]; [25]). We hypothesized that by bringing MBC principles to bear on vigilance, significant reductions in vigilance-related behaviors would be ensured. We also hypothesized that the SBAF-PTSD would be associated with treatment outcomes.

#### 6.1.2. Participants

Participants were 66 veterans who completed Prolonged Exposure Therapy (PE) or Trauma Processing and Safety Behavior Elimination (TP-SBE) (formerly known as Behavior Therapy for Anxiety and PTSD). All participants were diagnosed with PTSD. Twenty-four (36.4%) participants completed PE, and forty-two (63.6%) completed TP-SBE. All participants were male, except for 4 females, and the mean age was 48.8 years (*SD* = 14.5). Combat was the most common index trauma (*n* = 37; 59.7%), followed by “other” (*n* = 11; 17.7%), aftermath exposure (*n* = 9; 14.5%), and sexual trauma (*n* = 5; 8.1%). Chi-square analyses revealed significant differences in trauma type between participants who received TP-SBE and PE. Specifically, TP-SBE participants were more likely to have experienced combat trauma (*chi-square* 8.0, *df* = 1, *p* = 0.005). All participants were diagnosed with PTSD. Twenty-five carried comorbid diagnoses of depression (37%) and 19 had a comorbid anxiety disorder (28%).

#### 6.1.3. Treatments and Therapists

*Prolonged Exposure Therapy (PE)*. PE is a behavioral-based treatment designed to address symptoms of trauma-related avoidance and intrusions. The primary interventions include (1) psychoeducation; (2) breathing retraining; (3) in vivo exposure, and 4) imaginal exposure. In vivo exposure typically begins in session two and is largely carried out through homework assignments. Imaginal exposure typically begins in session three and continues through the final session.

*Trauma Processing and Safety Behavior Elimination (TP-SBE).* Trauma Processing and Safety Behavior Elimination (TP-SBE) is a treatment that focuses on processing trauma and eliminating PTSD-related safety behaviors. The treatment has three phases: (1) Preparing for Treatment; (2) Trauma Processing; and (3) Countering Vigilance and other Safety Behaviors. The first phase consists of socialization to treatment, education about PTSD and safety behaviors, identifying patient-relevant safety behaviors, and providing the rationale for treatment. The second phase of treatment (Processing Trauma) focuses on processing trauma memories while also countering suppression–distraction–internal avoidance-related safety behaviors. The third phase of treatment (Countering Vigilance and Other Safety Behaviors) begins with countering vigilance exercises and then moves into addressing other trauma-related safety behaviors. TP-SBE also has an optional phase with three additional modules: (1) Addressing trust-related safety behaviors; (2) countering withdrawal and avoidance safety behaviors; and (3) addressing rumination.

All cases were treated (*n* = 42) or supervised (*n* = 24) by the first author (JTG). Supervised cases were completed by 12 therapists-in-training, including 2 postdoctoral fellows, 7 pre-doctoral interns, and 3 pre-doctoral practicum students. The breakdown of cases completed by therapists-in-training were as follows: 4 cases were completed by postdoctoral fellows, 16 cases were completed by pre-doctoral interns, and four were completed by pre-doctoral practicum students. All therapists-in-training attended a 3-day PE training and a 1-day TP-SBE training provided by the first author. All cases were discussed in weekly supervision. Analysis of Variance found no significant difference in PCL-Difference scores by clinician status (Licensed Provider by Trainee) (*F* = 1.3, *df* = 1, *p* = 0.25).

#### 6.1.4. Measures

*SBAF-PTSD Scale.* See description from Study 2.

*PCL-5.* See description from Study 1.

Procedure

Participants attended an individual treatment planning session which detailed the treatments offered through the PCT team. Participants in the current study selected either PE or TP-SBE as their treatment of choice. Participants were administered the SBAF-PTSD and PCL-5 along with several other measures at pre- and post-treatment. Additionally, the SBAF-PTSD and PCL-5 were administered repeatedly during treatment (every 1–3 sessions).

#### 6.1.5. Data Analytic Procedures

The analytic strategy is sequential in nature, involving a hierarchical multiple regression approach conducted in R (Version 4.2.3). We first established a linear regression model with age, baseline SBAF-PTSD, and baseline PCL-5 (i.e., pre-treatment) scores as covariates. We then added our primary variable of interest, the change in SBAF-PTSD from pre- to post-treatment (Δ SBAF-PTSD), to the second regression model. The third regression model included the above predictors, as well as treatment condition (BTAP coded as 0; PE coded as (1)) to assess for differences between therapies. Unstandardized regression coefficients, standard errors, p-values for each predictor variable, and multiple R^2^ are reported for each model. To assess the incremental contribution of our primary variable of interest and the relevance of treatment condition, we compared the models sequentially using an Analysis of Variance (ANOVA) approach, which enabled us to examine whether the inclusion of Δ SBAF-PTSD or treatment condition significantly improved model fit.

### 6.2. Study 4—Results

The mean pre-treatment score on the SBAF-PTSD was 20.8 (*SD* = 4.95) and for the PCL-5, the mean pre-treatment score was 46.1 (*SD* = 11.2). Participants in TP-SBE received an average of 9.3 (*SD* = 3.37) sessions, closely matched by those in PE with an average of 10.2 (*SD* = 2.16) sessions. Post-treatment, the mean SBAF-PTSD score reduced to 12.4 (*SD* = 6.3), and PCL-5 score decreased to 20.6 (*SD* = 11.9), resulting in an average decrease of 8.8 (*SD* = 5.04) for SBAF-PTSD and 25.6 (*SD* = 13.5) for PCL-5 scores.

The first regression model included age (β = −0.04, *SE* = 0.11, *p* = 0.708), baseline SBAF-PTSD (β = 0.42, *SE* = 0.28, *p* = 0.146), and baseline PCL-5 (β = −0.51, *SE* = 0.15, *p* = 0.002) as covariates. These predictors accounted for 22% of the variance in ΔPCL-5, *p* = 0.016. In the second model, the primary variable of interest, change in SBAF-PTSD (ΔSBAF-PTSD) from pre- to post-treatment, was added. Inclusion of ΔSBAF-PTSD (β = 1.16, *SE* = 0.25, *p* < 0.001) significantly improved its explanatory power, *p* < 0.001, accounting for an additional 29% of the variance in ΔPCL-5. A third model was run that added treatment condition as an additional predictor. However, including treatment condition (β = 0.65, *SE* = 2.92, *p* = 0.825) did not significantly improve model fit, *p* = 0.83, suggesting ΔSBAF-PTSD was central to ΔPCL-5 regardless of treatment condition.

### 6.3. Study 4—Discussion

The purpose of Study 4 was to test the clinical utility of the SBAF-PTSD. As predicted, results showed that changes in the SBAF-PTSD scale were predictive of PTSD outcomes in both treatment conditions. This suggests that using the SBAF-PTSD scale in PTSD treatments may have clinical utility as it accounts for variance in treatment outcomes.

## 7. Discussion

Safety behaviors are consistently associated with the development and maintenance of anxiety symptoms and disorders. But most of the research in this area has focused on safety behaviors related to social anxiety. This is a problem because theory indicates that safety behaviors should also be involved with PTSD outcomes. One reason for the paucity of research in PTSD is the lack of a reliable, valid, and easy to use measure for assessing PTSD-specific safety behaviors. Developing such a measure is important for determining the degree to which specific kinds of PTSD-related safety behaviors serve as risk factors for PTSD as well as for treatment monitoring. The purpose of the current research was to publish a measure that meets this need.

We created the SBAF-PTSD scale and tested it in four studies. The scale was created by first selecting twelve items from the full-scale SBAF based on their associations with PTSD symptoms. Then we conducted a series of CFAs and reliability tests that led us to delete two items, resulting in the final 10-item SBAF-PTSD scale (see Appendix A). The results of three studies showed the scale to assess a unitary construct of PTSD-related safety behaviors and a three-item Social Escape Index. The scale demonstrated strong reliability, convergent and divergent validity, and predictive validity. Given the procedures for item selection, one might question how this measure differs from the full-scale SBAF. We contend it is a shorter, specialized version of the SBAF. If time is not a concern, then it would be possible to administer the entire SBAF and simply select the 10 items we identified to create an SBAF-PTSD scale score. This option might be useful in research where time is not limited, and the researcher is interested in information about a broad range of safety behavior use. However, if the researcher is primarily concerned with PTSD, then the short scale would be the better option. Similarly, for clinicians interested in a short, easy to administer measure that is associated with PTSD treatment outcomes, the SBAF-PTSD scale would be the most feasible option.

The results of Study 4 were particularly encouraging for the usefulness of the SBAF-PTSD in real-world clinical settings (in this case a VA hospital). Results showed the measure could predict outcomes in a PTSD effectiveness treatment study. In this study, the SBAF-PTSD scale was used throughout treatment as part of measurement-based care (i.e., treatment monitoring). The resulting effect for the treatment groups was an approximate 10-point reduction on the scale, suggesting large reductions in vigilance-related behaviors. This is a promising finding, as vigilance has been shown to be resistant to change in past PTSD treatment studies and the SBAF-PTSD scale provides a tool for assessing progress.

The research had several strengths. First, we tested the new measure in four studies. This allowed us to test the measure and replicate findings in a variety of settings using both clinical and non-clinical populations. Further, we conducted a strong test of the measure by assessing several psychometric properties including factor structure, internal consistency, inter-item correlations, divergent validity, convergent validity, and predictive validity. Finally, we used the measure in a real-world treatment setting, demonstrating its potential clinical utility.

There were also limitations. For example, three of the four studies used cross-sectional designs, which did not allow us to disentangle temporal associations among variables. Thus, future research should use longitudinal designs to determine the degree to which PTSD-specific safety behaviors confer risk for the prospective development of PTSD. Additionally, the participant samples were somewhat homogenous with regards to race/ethnicity and especially geography (all four studies used samples from the United States). Thus, the next step in the validation of this measure and research on PTSD-specific safety behaviors is to conduct studies with diverse samples from around the globe. Finally, future research should attempt to establish normative standards. It would be useful for researchers and clinicians to have information that would allow them to determine who is using safety behaviors at a level that may confer risk for psychopathology or harm treatment outcomes.

In conclusion, we consider the SBAF-PTSD scale from a construct validity perspective. According this this framework ([9]), a measure is judged by how it behaves in accordance with theory. In this case, the question is whether the new measure of PTSD-specific safety behaviors behaves like a measure of PTSD-specific safety behaviors is supposed to behave according to the theories (i.e., the nomological net). First, a measure of PTSD-specific safety behaviors should be associated with symptoms of PTSD. Results of all four studies supported this. Second, a measure of PTSD-specific safety behaviors should not be as strongly associated with general measures of well-being as it is with PTSD symptoms. Results of Study 2 supported this. Third, a measure of PTSD-specific safety behaviors should also be associated with precursors and concomitants of PTSD (such as intrusive thoughts and repetitive thinking). Study 2 supported this assertion. Further, a measure of PTSD-specific safety behaviors should be able to differentiate people with varying levels of PTSD (i.e., predict group status). This was supported in Study 3. Finally, a measure of PTSD-specific safety behaviors should be sensitive to changes in safety behavior use and, ideally, predict treatment outcomes (assuming a causal or maintenance role in PTSD). Study 4 supported this hypothesis. Taken together, the results provide preliminary evidence of construct validity (in addition to traditional metrics of reliability and validity) in both clinical and non-clinical samples. An effectiveness study also suggests that it may have clinical utility. By administering the SBAF-PTSD repeatedly during treatment (i.e., treatment monitoring), it may be possible to predict treatment gains and end-of-therapy outcomes. We look forward to continued research testing the predictive validity and clinical utility of the SBAF-PTSD scale.

## Figures and Tables

**Figure 1 behavsci-15-01248-f001:**
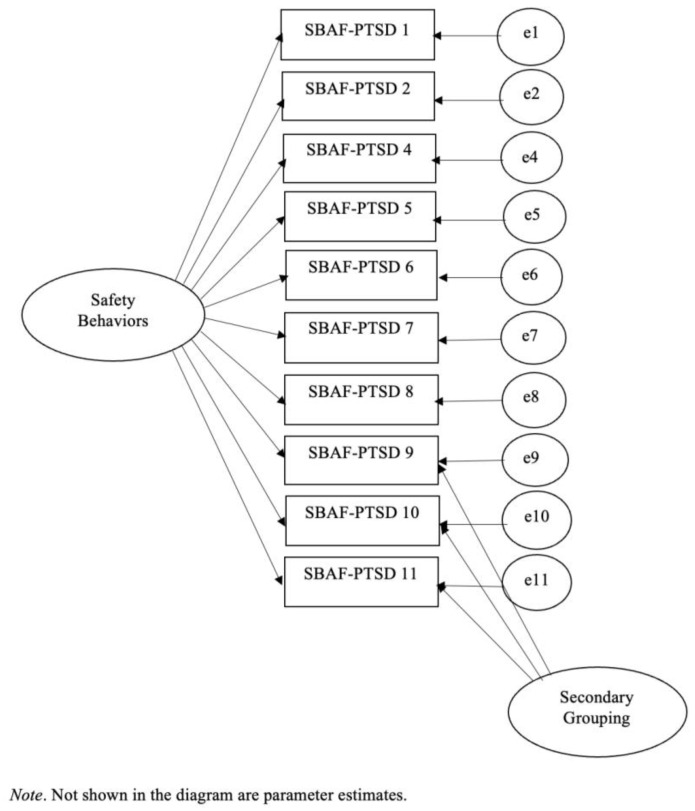
CFA model of the bi-factor SBAF-PTSD.

**Figure 2 behavsci-15-01248-f002:**
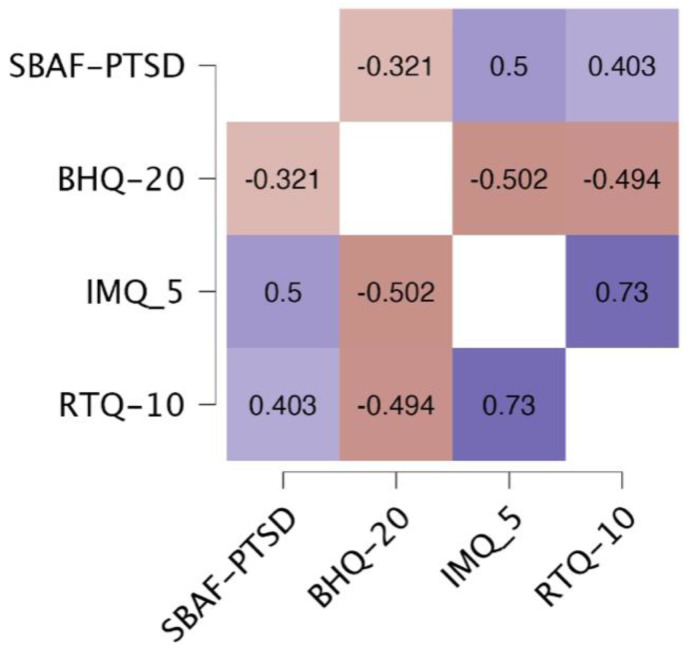
Correlations among the SBAF-PTSD, BHQ-20, IMQ-5, and RTQ-10 (all correlations significant *p* < 0.001).

**Table 1 behavsci-15-01248-t001:** SBAF-PTSD Items, Means, Standard Deviations, and Prior Theoretical Designations.

Item	Content	*M*	*SD*	P vs. R	V vs. S
1	Scope places out before entering.	2.17	0.87	P	V
2	Sit with back to wall.	2.40	0.81	P	V
3	Rush through the stores or go directly to desired items and leave as quickly as possible.	2.07	0.98	P	-
4	Check yard or the area around your home (“Perimeter Checks”).	1.85	1.00	P	V
5	Make up contingency plans in case someone is physically aggressive or there is some kind of emergency.	2.28	0.89	P	V
6	Walk slowly to let someone pass who is close behind.	2.04	0.94	R	S
7	Watch others for signs of danger.	2.39	0.77	P	V
8	Check locks on doors or windows.	2.43	0.83	P	V
9	Pretend I do not see or recognize someone so that I do not have to speak with them.	1.58	0.89	P	S
10	Cut conversation short.	1.76	0.78	R	S
11	Leave events or activities early.	2.01	0.88	R	S
12	Stay on the outside of crowds and/or monitor for exits or escape routes.	2.30	0.83	P	V

Note. P = preventative safety behaviors; R = restorative safety behaviors; V = vigilance; S social; (-) did not load on any factor in previous studies.

**Table 2 behavsci-15-01248-t002:** Scale Descriptive Statistics and Internal Consistencies for the SBAF-PTSD, the PHQ-9, and the PCL-5.

*Scale*	*M*	*SD*	*Score Range*	*Alpha*
SBAF–PTSD 10	20.87	5.54	0–30	0.84
SBAF–PTSD Social	5.34	2.15	0–9	0.80
PHQ–9	15.63	6.20	0–27	0.86
PCL–5	52.91	14.33	9–71	0.91
Intrusions	12.81	4.44	2–20	0.85
Avoidance	5.97	1.87	0–8	0.77
Cognitions and Mood	18.06	6.11	0–28	0.81
Arousal	15.99	4.65	1–24	0.76

Note. SBAF–PTSD = Safety Behavior Assessment Form–PTSD; For the SBAF–PTSD, statistics for the original measure are reported first, and statistics for the derived scales follow. Scores based on N = 169 for the PHQ–9 and the SBAF–PTSD scales and N = 165–167 for the PCL–5 scales.

**Table 3 behavsci-15-01248-t003:** Means and Standard Deviations for SBAF-PTSD as a Function of Trauma Grouping.

	No Trauma Mean (SD)	Trauma/Minimal Sxs Mean (SD)	Trauma/Probable PTSD Mean (SD)
PTSD–SBAF Scale	11.2 (4.5)	11.1 (4.8)	13.8 (4.3)

## Data Availability

Data for Studies 1 and 4 are not available due to regulations of the VA Health Care System. Data for Study 3 are available upon request. Data for Study 4 are available here: https://osf.io/vygse/ (accessed on 26 June 2025).

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
