# Peer review of "Development and Validation of the Safety Behavior Assessment Form-PTSD Scale"

_behavsci, 2025, doi:10.3390/bs15091248_

Round 1
Reviewer 1 Report
Comments and Suggestions for Authors
p2: please introduce some examples for PTSD safety behaviors as was presented for SAD and GAD
p2: please clarify "trauma-related anxiety" in the SBAF. that sounds like PTSD to me.
p2: the purpose of identifying a subscale from the SBAF (versus creating a new scale) should be clearer throughout the manuscript (e.g., abstract)
p2: given that the SBAF is the primary focus of this paper, the scale development, psychometric properties, and past factor analytic investigations should be reviewed in detail.
p2: does the SBAF have a trauma-related anxiety subscale? how does that compare in content and length to what is proposed here?
p3: how was the sample size determined for the study/CFA?
p3: how was diagnostic status determined? CAPS?
p3: why not use an EFA (or CFA) design for the full SBAF?
p3: what were the criteria for item selection? a correlation of .21 doesn't sound particularly impressive given the context. Were they the 12 items with the highest correlations, or were there other factors that contributed (e.g., review of item content by PTSD experts)?
p4: relatedly, where did the preventative and restorative factors and vigilance and social come from? previous EFAs?
p4-5: the CFA plan/analyses are messy/difficult to follow rationale/justification.
p4-5: it seems like this study skipped a step, or is omitting a previous step that lead to these CFAs. relatedly, why is the structure of the new subscale needed? i can understand needing to identify/separate the items from the full SBAF scale (which was skipped here). But, once separated, the items predictive ability/fit with PTSD symptoms seems more interesting.
p6: what analyses were used to support the suggest cutscore of 19? curve analyses?
p7: the first line of the discussion is the critical item here. the authors should focus more on that process and how the unpublished data was used to support those decisions. it's the 12 items (from the 41 full scale) that is most interesting, rather than the questions regarding the fit/structure of the 12/10 items post-selection.
p7: what test/data supports the label of the Social Escape Index (convergent/discriminant validity)?
p8: i wouldn't associate item 8 with PTSD (definitely SAD). Items 7 and 9 are iffy too.
p9: how was the sample size of 142 selected for the CFA analyses?
p9: please provide more detail on the use of the IMQ-5 and RTQ-10 in patients with PTSD. are they PTSD measures, or measures of general symptoms that happen to correlate highly with PTSD?
p11: how was the sample size of 150 selected for the 3rd study?
p11: please provide more detail on the demographics of this sample, and for other samples (age, sex, etc).
p11: the rate of trauma exposure appears quite low.
p11: has the PC-PTSD-5 been used to separate patients into 3 groups in the past? please cite. i'm not sure of the utility of the 2 vs more symptom groups.
p14: i could not locate Table 4
p15: based on the methods and sample sizes, i would be more tentative in the descriptions of the findings (e.g., provide initial support vs. psychometrically sound and a viable tool).
Author Response
Please see attached attachment.

Reviewer 2 Report
Comments and Suggestions for Authors
- Its not clear why the authors chose to begin the initial validation process by narrowing the original SPAF down to 12 items and investigate only these 12 items in this paper. Without understanding the correlations of these 12 items in the original sample (N = 115 treatment seeking veterans), I have concerns that the authors may have missed something and therefore developed an instrument that is not generalizable enough.
- In study 1 the reporting of the development of the latent profile of the scale switched tenses (from past to present) and was entirely too detailed to make sense of the procedure and results.
- It is not entirely surprising that there were no demographic associations to the Study 1 scale given that the Study 1 sample was largely homogeneous in terms of white, middle-aged, and male. Thus, further validation needs to occur in more diverse samples.
- More information would be helpful in Study 1 about sample characteristics, specifically what the authors mean by “20% reported as another race” and “trauma/stressor disorder”. These things are not clear and yet are important characteristics about the sample.
- In study 2, without information about race and ethnicity, it’s difficult to make any conclusions about how this provides validation. Particularly concerning here is that the authors do not report the full demographic composition of their expected sample – the race/ethnicity numbers here only add up to 85%. What is the other 15% of the expected population?
- In Study 2 the authors state that the scoring of the SBAF-PTSD ranges from 0-36 yet the scale is 10 points – thus, wouldn’t the total score possible be 30 instead of 36?
- For study 3 the authors aimed to test if the SBAF-PTSD could predict change in PTSD symptom severity following treatment. However, the original design of the SBAF-PTSD was established based on items highly correlated with PCL items; thus, this finding is not surprising. The findings establish that a measure highly correlated with PTSD accounts for change in PTSD symptoms following treatment.
- The discussion lacked information on which items were selected – it would be helpful to understand what the items mean and how the measure if unique from the original SBAF.
Round 2
Reviewer 1 Report
Comments and Suggestions for Authors
I was impressed by the authors' work to address my original critiques. I do not identify any additional significant concerns and do not have any additional feedback.
Reviewer 2 Report
Comments and Suggestions for Authors
The authors have sufficiently addressed my concerns